# Gut microbiota diversity in human strongyloidiasis differs little in two different regions in endemic areas of Thailand

Rutchanee Rodpai[1,2], Oranuch Sanpool[1,2], Penchom Janwan[2,3], Patcharaporn Boonroumkaew[1,2], Lakkhana Sadaow[1,2], Tongjit Thanchomnang[2,4], Pewpan M. Intapan[1,2], Wanchai Maleewong[1,2]*

1 Department of Parasitology, Faculty of Medicine, Khon Kaen University, Khon Kaen, Thailand, 2 Mekong Health Science Research Institute, Khon Kaen University, Khon Kaen, Thailand, 3 Department of Medical Technology, School of Allied Health Sciences, Walailak University, Nakhon Si Thammarat, Thailand, 4 Faculty of Medicine, Mahasarakham University, Mahasarakham, Thailand

* wanch_ma@kku.ac.th

**Data Availability Statement:** All sequence reads have been deposited at the NCBI Sequence Read Archive (SRA) under project accession number

## Abstract

Human gastrointestinal helminthic infections have a direct and/or indirect effect on the composition of the host gut microbial flora. Here, we investigated the effect of infection with a soil-transmitted intestinal nematode, *Strongyloides stercoralis*, on the gut microbiota of the human host. We also investigated whether composition of the microbiota in infected persons might vary across endemic regions. Fecal samples were obtained from volunteers from two areas endemic for strongyloidiasis, Khon Kaen Province in northeastern Thailand and Nakhon Si Thammarat Province in southern Thailand. Samples from Khon Kaen were from infected (SsNE) and uninfected (NegNE) individuals. Similarly, samples from the latter province were from infected (SsST) and uninfected (NegST) individuals. DNA sequences of the V3-V4 regions of the bacterial 16S rRNA gene were obtained from the fecal samples. No statistical difference in alpha diversity between groups in terms of richness or diversity were found. Statistical difference in beta diversity was observed only between NegNE and NegST. Some significant differences in species abundance were noted between geographical isolates. The SsNE group had a higher abundance of *Tetragenococcus holophilus* than did the SsST group, whereas *Bradyrhizobium* sp. was less abundant in the SsNE than the SsST group. For the uninfected groups, the NegNE had a higher abundance of *T. holophilus* than the NegST group. Our data showed that *S. stercoralis* infection leads to only minor alterations in the relative abundance of individual bacterial species in the human gut: no detectable effect was observed on community structure and diversity.

## Introduction

The microbiota in the human gastrointestinal (GI) tract includes more than 1000 species of bacteria that play essential roles in human health, including nutrient metabolism, protection against pathogens and regulation of immune responses [1, 2]. Dysbiosis of the gut microbiota

PRJNA807511. Please see link: https://www.ncbi.nlm.nih.gov/bioproject/PRJNA807511.

**Funding:** This study was supported by the National Science, Research and Innovation Fund (NSRF), Thailand for Khon Kaen University, a grant from Research Program, Research and Graduate studies, Khon Kaen University (OS, PMI and WM; grant number RP-65-3-001), a Scholarship under the Post-Doctoral Training Program from Khon Kaen University (RR; grant number PD2565-02-03), and grant from the Faculty of Medicine, Khon Kaen University (WM and OS, grant number RG63301). The funders had no role in study design, data collection and interpretation, or the decision to submit the work for publication.

**Competing interests:** The authors have declared that no competing interests exist.

is linked to human diseases such as inflammatory bowel disease (IBD), irritable bowel syndrome (IBS), metabolic syndrome and cardiovascular disease [3–5]. Multicellular organisms occurring in the GI tract, such as parasitic worms or helminths, are also considered to be harmful to human health [6, 7]. The commensal microbiota has coevolved to share the host environment as much with helminths [8, 9] as with the mammalian hosts themselves [10]. The association between soil-transmitted helminth (STH) infections and gut microbiota diversity and composition has been the subject of several recent investigations, with diverse and contrasting results [7, 11–13].

*Strongyloides stercoralis* is a soil-transmitted intestinal nematode estimated to infect at least 600 million individuals worldwide [14, 15]. Many cases of infection with *S. stercoralis* are mild or asymptomatic [6]. In Thailand, *S. stercoralis* is highly prevalent in several regions of the country and infection is recognized as a medical problem. Recently, the prevalence of *S. stercoralis* infections in the northern, central, northeastern, and southern regions of Thailand was estimated at 15.9% [16], 2.47% [17], 22.8–23.0% [18, 19] and 0.5–0.9% [20, 21], respectively. These differences in *S. stercoralis* prevalence between regions may be due to differences in host age, season of observation, parasitological technique utilized, personal hygiene habits and cultural, socioeconomic and environmental factors [20, 22, 23]. The parasitic adult stages of *S. stercoralis* inhabit the GI tract of humans and other vertebrates [24], where their interactions with the host gut microbiota can impact, and may alter, the host gut microbiome [7, 11, 12]. Furthermore, race and geographic location may influence the composition and diversity of the gut microbiota [25]. The southern region of Thailand differs the most in terms of geography, population behavior and diets from the northeastern region [26, 27]. In addition, variations in gut microbiota may be altered in *S. stercoralis* infection. In the present study, we investigated the effect of *S. stercoralis* infection on the composition of human gut microbiota and whether such effects differed between endemic regions.

## Materials and methods

### Ethics statement

The study protocol was approved by the Khon Kaen University Ethics Committee for Human Research (HE611059) with relevant guidelines and regulations covering Ethical Principles for Medical Research Involving Human Subjects by the Declaration of Helsinki. Written, informed consent was obtained from adult participants and from the parents or legal guardians of minors.

### Fecal samples from individuals with or without strongyloidiasis

Approximately 40 grams of each fecal sample were obtained from 10 volunteers infected or not with *S. stercoralis* from Khon Kaen Province in northeastern Thailand. Similar 10 samples were obtained from Nakhon Si Thammarat Province in southern Thailand. The volunteers live in their localities; never travel between both areas. All participants in this investigation were healthy, and the individual *S. stercoralis* infections were asymptomatic strongyloidiasis. The specimens were examined for parasites by the agar plate culture method [28] and by the formalin-ethyl acetate concentration method [29]. Aliquoted 200 mg of each stool sample was preserved in 75% alcohol for transfer to the laboratory. The stool samples were categorized in four groups. Groups were free of all parasitic infections ("Neg. . .") or infected only with *S. stercoralis* ("Ss. . ."). The groups from northeastern Thailand (Khon Kaen Province) were therefore labelled "NegNE" (N = 5; male = 3, female = 2, Age [range 49–77, mean 59.2 years]) and "SsNE" (N = 5; male = 2, female = 3, Age [range 45–74, mean 57.4 years]). Those from southern Thailand (Nakhon Si Thammarat Province) were labelled "NegST" (N = 5; Male = 3,

Female = 2, Age [range 23–70, mean 53.4 years]) and "SsST" (N = 5; Male = 3, Female = 2, Age [range 25–67 years, mean 51 years]). Each stool aliquot was frozen at -20˚C to minimize further microbial activity until use for DNA preparation.

## Bacterial DNA extraction and sequencing

Each total DNA was isolated from each fecal sample (200 mg per sample) using a QIAamp DNA stool mini kit (Qiagen, Hilden, Germany) following the manufacturer's protocol. Briefly, the fecal samples were extracted in optimized buffers in combination with inhibitEX buffer (Qiagen) to remove PCR inhibitors. Proteinase K was subsequently added. DNA from lysates was isolated using a DNA-binding silica-gel membrane column. Remaining impurities were removed in two wash steps. Amplification-ready DNA was then eluted in low-salt buffer (Qiagen). Sample quality control was performed by monitoring the DNA concentration and purity using a NanoDrop Spectrophotometer (Thermo Fisher Scientific, Waltham, MA), electrophoresis through 1% agarose gels, and use of the Qubit 2.0 fluorometer (Thermo Fischer Scientific, Australia). DNA was diluted to 1 ng/µL using sterile water and used to analyze individually. These processes were performed on the same day when the sample was firstly thawed, except for SsST samples no. 3–5 were performed in another batch.

The V3-V4 regions of the 16S rRNA gene were amplified using universal region-specific primers 341F (5'-CCT AYG GGR BGC ASC AG-3') and 806R (5'-GGA CTA CNN GGG TAT CTA AT-3') (Novogene, Singapore) connecting with sample-identifying barcodes. All PCR reactions were carried out with Phusion® High-Fidelity PCR Master Mix (New England Biolabs, Ipswich, MA). The DNA libraries was generated based on an IonS5™XL (Thermo Fisher Scientific) (NegNE no. 1–5, SsNE no. 1–5, NegST no. 1–5, and SsST no.1-2) and the Illumina Novaseq 6000 platform (Illumina, USA) (SsST no. 3–5), and quantified via Qubit and Q-PCR. Single-end and paired-end raw reads were generated. All sequence reads have been deposited at the NCBI Sequence Read Archive (SRA) under project accession number PRJNA807511.

## Bioinformatics analyses

Raw reads were assigned to samples based on their unique barcode and truncated by cutting off the barcode and primer sequence. Quality filtering on the raw reads was done under specific filtering conditions to obtain high-quality clean reads [30] using QIIME (V1.7.0, http://qiime.org/scripts/split_libraries_fastq.html) [31]. The reads were compared with the reference database (Gold database) using the UCHIME algorithm (http://www.drive5.com/usearch/manual/uchime_algo.html) to detect chimeric sequences [32], which were removed [33]. The remaining reads were effective reads.

Clustering of operational taxonomic units (OTUs) from all the effective reads was performed using Uparse software (V7.0.1001, http://drive5.com/uparse/) [34]. Sequences with ≥ 97% similarity were assigned to the same OTU. The OTU of the individual samples was divided into four groups, and the mean abundances of each OTU were computed for further analysis. A representative sequence for each OTU was screened for further annotation. Mothur software was used to interrogate the SSU rRNA database of the SILVA database (http://www.arb-silva.de/) for species annotation at each taxonomic rank (kingdom, phylum, class, order, family, genus, species) [35, 36].

Microbial community diversity and dissimilarity (alpha and beta diversity) were analyzed using QIIME (Version 1.7.0) and graphically displayed using R software (Version 2.15.3). Alpha diversity refers to species diversity using six indices, including observed-species, Chao1, Shannon, Simpson, ACE, Good's-coverage. Beta diversity based on both a weighted and an

unweighted UniFrac distance matrix was calculated. *T*-tests were used for analysis of significant differences between groups ($p \leq 0.05$). Differences in genera abundance between infected (Ss) and uninfected (Neg) groups, and between locations (NE and ST) were assessed with 2-way ANOVA using STAMP software [37].

## Results

### Microbial composition of gut microbiota in *S. stercoralis* infection

Basic statistics of the numbers of reads and OTUs in each sample are shown in S1 Table. A total of 4761 OTUs in 38 phyla were identified from the 4 sample groups (S2 Table). Firmicutes was the predominant phylum in all sample groups except for the NegNE group, in which individual samples varied substantially (Fig 1A). At the family level (Fig 1B), Lachnospiraceae

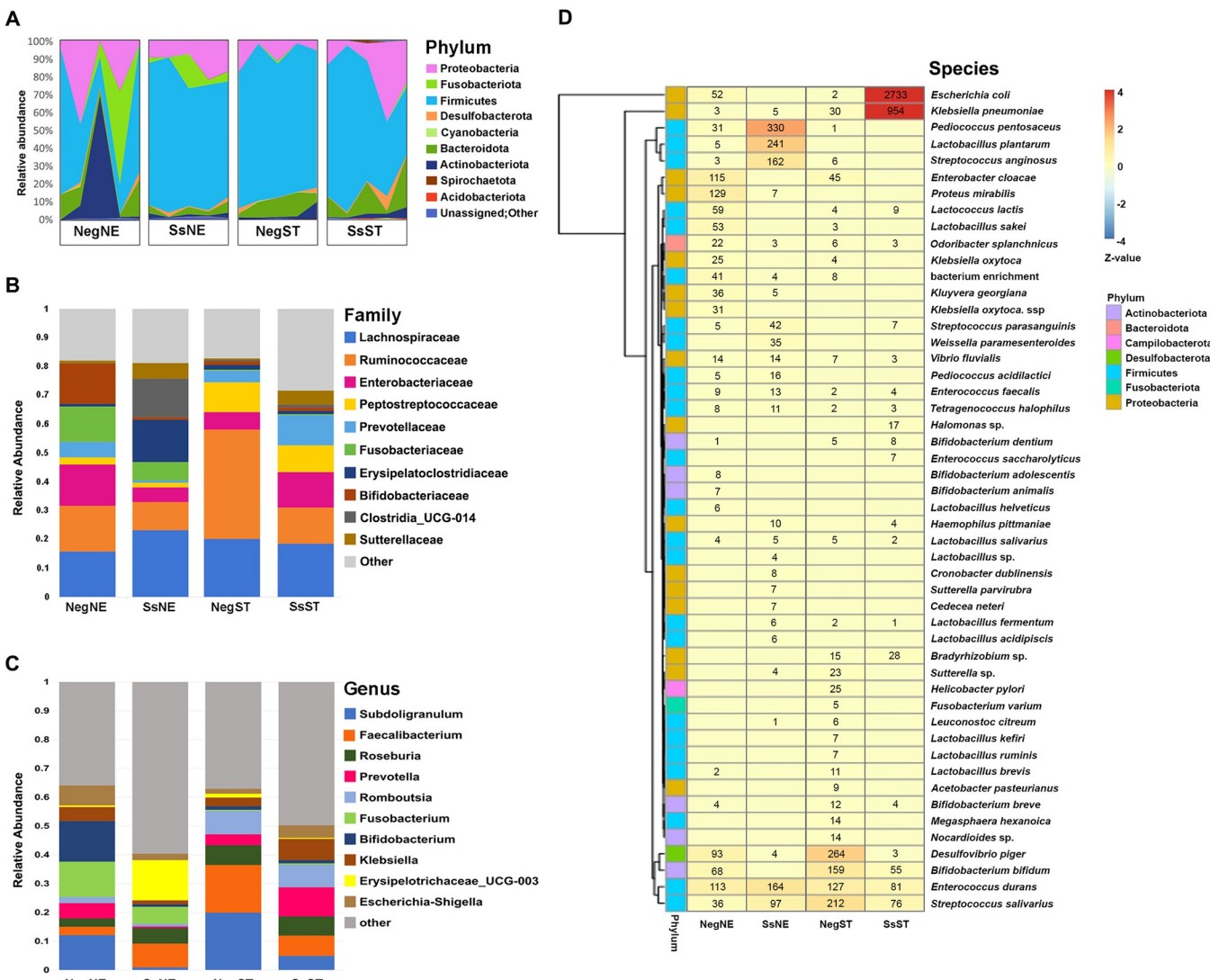

**Fig 1. Species distribution of the gut microbiota of two endemic areas of strongyloidiasis in terms of the most abundant.** (A) Area graph showing relative proportions of the most abundant phyla (n = 5 in each group). (B) The top 10 families and (C) genera in terms of relative abundance in each sample group. (D) Hierarchical clustering heatmap of dominant species plotted by the absolute z-value. The colors in the heatmap refer to the species abundance, according to the color bar on the right. Numbers in the heatmap are reads per sample.

was abundant in all sample groups. Ruminococcaceae was the predominant family in the NegST group, whereas the SsST group had approximately equal representation of Lachnospiraceae, Ruminococcaceae, Enterobacteriaceae, Peptostreptococcaceae, and Prevotellaceae. The NegNE group also had a variety of families roughly equally represented including Lachnospiraceae, Ruminococcaceae, Enterobacteriaceae, Fusobacteriaceae and Bifidobacteriaceae. For the SsNE, the top three families were Lachnospiraceae, Erysipelatoclostridiaceae and Clostridia_UCG-014. The genus-level annotation is shown in Fig 1C. The top 10 genera varied among sample groups. For the NegNE, the genera *Subdoligranulum*, *Fusobacterium* and *Bifidobacterium* were the most abundant. Genera *Erysipelotrichaceae*_UCG-003 and *Faecalibacterium* were abundant in the SsNE group. In the NegST group, *Subdoligranulum* and *Faecalibacterium* were predominant. For the SsST group, the wide range of genera represented included *Subdoligranulum*, *Faecalibacterium*, *Roseburia*, *Prevotella*, *Romboutsia*, *Klebsiella* and *Escherichia-Shigella* (Fig 1C). The dominant 50 species identified among all sample groups is displayed in the species-abundance heatmap (Fig 1D). Each group contained a few dominant bacterial species.

To determine the extent to which OTUs were shared between different groups, a Venn diagram was constructed (Fig 2). The number of OTUs unique to NegNE, SsNE, NegST and SsST were 876, 921, 1000 and 1403, respectively. These unique OTUs represented 68.60%, 74.09%, 69.93% and 77.47% of total OTUs in each of the four groups, respectively. OTUs shared between groups were investigated. Thirty-nine OTUs were shared in samples from strongyloidiasis individuals from northeastern and southern regions (SsNE vs SsST), and 52 OTUs were shared in the negative groups (NegNE and NegST). Within the same region, 38 OTUs were shared by the NegNE and SsNE groups and 77 by the NegST and SsST groups. The number of core OTUs, present in all groups, was 147. These core OTUs belonged to 35 families, of which Lachnospiraceae was the most abundant (52 OTUs) (Fig 2 and S3 Table).

## Bacterial community diversity and similarity between regions and strongyloidiasis status

To assess the species diversity of the bacterial communities, several comparisons were performed using alpha and beta diversity metrics. Gut microbiota diversity in each group was indicated by alpha diversity indices. There was no statistical difference between groups regarding to the estimated richness (ACE, $p = 0.055$ and Chao1, $p = 0.055$) or diversity (Simpson, $p = 0.244$ and Shannon, $p = 0.348$) (Fig 3A). The differences between microbial communities were also analyzed in terms of beta-diversity. There was significant difference between NegNE and NegST ($p = 0.001$) in weighted UniFrac distances. There were no other significant differences in weighted UniFrac among other groups (Fig 3B). The dissimilarity coefficient between pairwise sample groups was plotted, in which the beta diversity ranged from 0.25 to 0.32 (Fig 3C).

## Microbial species differing between groups

Statistical pairwise analysis of different communities was performed using *t*-test analysis (S1 Fig). For SsNE and SsST group comparison, the SsNE group showed a significantly higher abundance of *Tetragenococcus holophilus*, which the mean value of the species abundance was 0.02% and 0.004%, respectively [$p = 0.012$; 95% confidence interval (CI) 0.001 to 0.025]. *Bradyrhizobium* sp. has significantly lower abundance in SsNE group than did the SsST group [$p = 0.041$; 95% CI -0.08 to 0.00]. For the comparison of non-infected groups, the NegNE group had a higher abundance of *T. holophilus* than the NegST group, which the mean value of the species abundance was 0.014% and 0.002%, respectively [$p = 0.033$; 95% CI 0.001 to

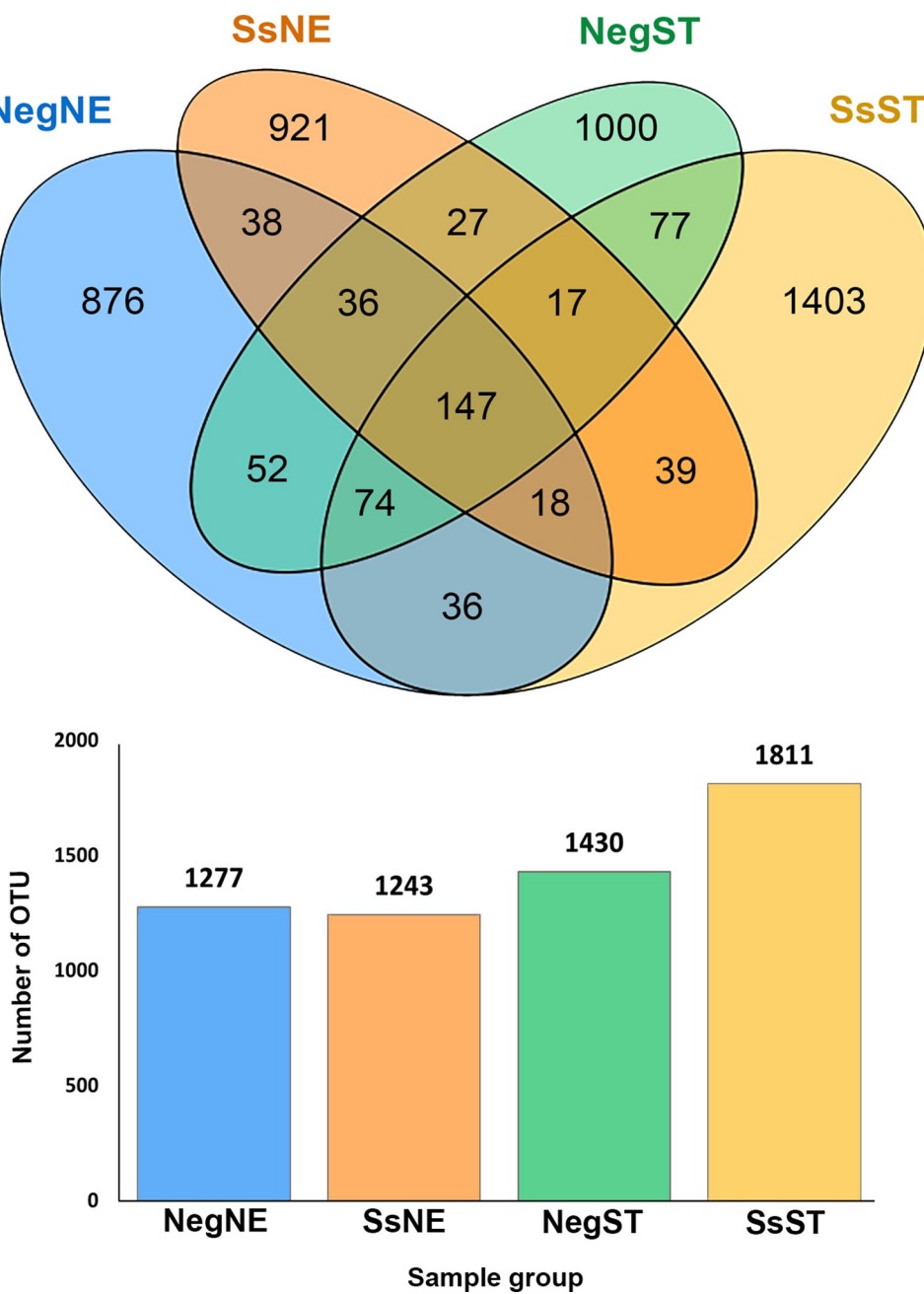

**Fig 2. Venn diagram showing numbers of unique, shared and core bacterial OTUs among sample groups.** Each ellipse represents one group. Values represent number of OTUs in each overlapping or unique segment. The lower panel graph represents total number of OTUs in each sample group.

0.022]. Two-way ANOVA was used to assess statistically significant differences in bacterial abundance at the genus level (S4 Table). Individual samples were divided into 4 groups: uninfected (NegNE and NegST), and infected (SsNE and SsST), northeastern (NE; SsNE and NegNE), and southern (ST; SsST and NegST) group. Statistic difference between the mean proportion abundance was found in few taxa among groups. In the comparison of uninfected and infected groups, the uninfected group had more *Ordoribacter* and *Allisonella* than the

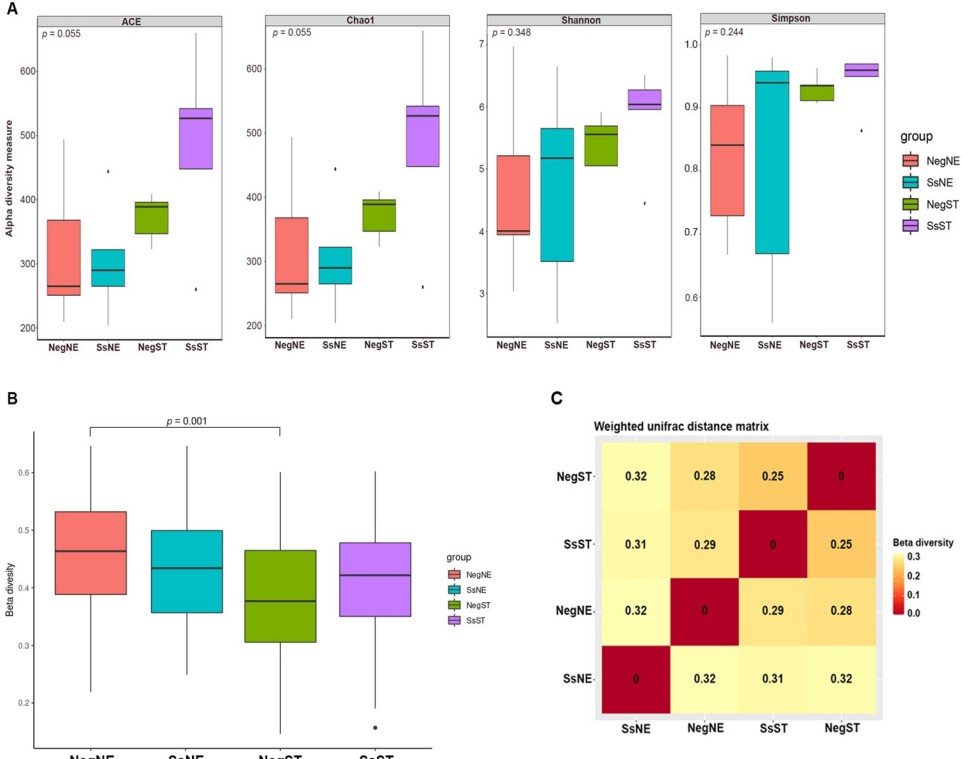

**Fig 3. Measures of gut microbiome diversity and similarity between regions in samples from individuals infected and uninfected with *Strongyloides*.** (A) Boxplot of alpha diversity indices. The ACE and Chao1 indices estimate the OTU richness in samples. Shannon and Simpson indices reflect the OTU diversity in samples. The greater the ACE or Chao1 indices, the higher expected species richness of the microbiome; the smaller the Simpson or the greater the Shannon, the higher the diversity of the microbiome. Boxes represent the interquartile range, lines indicate medians, and whiskers indicate the range. Wilcoxon rank-sum test and Tukey's test are used for statistically significant differences between groups ($p < 0.05$). (B) Boxplots based on weighted UniFrac distances to show the difference of beta diversity between groups (statistically significant differences between groups was set at $p < 0.05$). (C) Heatmap of the beta diversity matrix of the gut microbiota based on weighted UniFrac distances to reflect the dissimilarity between samples groups.

infected group ($p < 0.05$). Nonetheless, in a different geography comparison, there was a significant difference in the abundance of the genus *Tetragenococcus* in the NE group higher than the ST group. While, the genus *Bradyrhizobium* has higher abundant in the ST group than in the NE group.

## Discussion

The gut microbiota has an influence on human health and may response to disease [1, 3]. There is evidence that infection with gastrointestinal parasitic helminths can alter the host gut microbiome [7, 11, 12, 38].

In the present study, Firmicutes was the main phylum represented in the microbiota of *Strongyloides*-infected and uninfected samples. A similar finding has been reported for strongyloidiasis and other helminth infections [39, 40]. Firmicutes, Bacteroidetes and Actinobacteria are prominent among the gut biota of adult humans [41]. The families Lachnospiraceae and Ruminococcaceae co-dominate in fecal bacteria of healthy adults [41]. In this study, the Ruminococcaceae had a high relative abundance in uninfected groups (NegNE and NegST) and a low relative abundance in infected groups (SsNE and SsST). Ruminococcaceae is

beneficial to intestinal health because it produces butyrate and other short-chain fatty acids (SCFAs) [42]. It has been discovered that its abundance is reduced in various inflammatory bowel diseases, including ulcerative colitis and Crohn's disease [43]. Furthermore, serum SCFAs levels in *Strongyloides*-infected people have recently been revealed to be decreased [44]. An infection with the intestinal parasite *S. stercoralis* might be the cause of the Ruminococcaceae family reduction. Geographic contrasts at the microbiota genus level were apparent. The genus *Bifidobacterium* was less abundant but the genus *Erysipelotrichaceae* UCG-003 was more abundant in SsNE samples than in NegNE. This decrease of *Bifidobacterium* in infected individuals is consistent with prior observations in Thai children infected with soil-transmitted helminths [7]. Bifidobacteria are probiotic microorganisms that help to maintain a healthy balance between the various floras in different parts of the human gut and also have other health benefits [45, 46]. On the other hand, increased proportions of the Erysipelotrichaceae have deleterious consequences for human health which may be associated with metabolic disorder or energy metabolism [4, 47]. Furthermore, the presence of this family has been proposed as a microbial biomarker in individuals with certain types of irritable bowel syndrome (IBS) [48]. In samples from southern Thailand, *Subdoligranulum* and *Faecalibacterium* both dominate in the NegST group but not in the SsST group. A relative decrease of *Subdoligranulum* in the infected group was also observed in SsNE. The genus *Subdoligranulum* (family Ruminococcaceae) can produce butyrate and enhance the gut health by providing energy for host cells and maintaining gut barrier integrity [49, 50]. Recently, a markedly decreased abundance of *Subdoligranulum* was observed in allergic rhinitis, associated with the Th2-dominated immune response with increased levels of serum immunoglobulin E, compared with healthy individuals [51]. Interestingly, the genus *Subdoligranulum* may be a promising microbial biomarker and a probiotic candidate for reducing the harmful effects of strongyloidiasis. Another butyrate-producing taxon, the genus *Faecalibacterium*, has been associated with glucose and lipid metabolism that associated with more severe disease in human, such as non-alcoholic fatty liver, non-alcoholic steatohepatitis, and obesity [52, 53]. *Faecalibacterium* is also associated with inflammatory properties: patients with normal gut mucosa have a higher abundance of *Faecalibacterium* than those with inflamed mucosa. Furthermore, *Faecalibacterium* was more abundant in responders to inflammatory bowel-disease therapy than in non-responders at baseline [5, 54–56].

We discovered that, although each group had a large number of distinct bacterial OTUs, some species were shared between groups. *Haemophilus pittmaniae* was only detected in the *Strongyloides*-infected groups (SsNE and SsST). It is a gram-negative bacterium (family Pasteurellaceae) and is part of the normal oropharyngeal flora [57]. This organism may be involved in respiratory-tract infection in individuals with lung disease [58]. The presence of this microbial species in the gut microbiome might be associated with *S. stercoralis* infection, bearing in mind that the worm's life cycle involves migration via the respiratory system [59]. Some species were found only in one geographic region. For example, *Lactobacillus plantarum* and *Proteus mirabilis* were found only in the northeastern Thai samples (Fig 1D, S2 Table). The latter species, a commensal microbe in the intestine that can be easily separated from sewage or contaminated soils and water, was most abundant in the NegNE group. It is not a major invasive pathogen and only rarely causes infection in healthy people [60]. In the SsNE group, lactic acid bacteria such as *L. plantarum* and *Pediococcus pentosaceus* were abundant. These bacteria seem to have a beneficial effect on human health. In particular, *L. plantarum* has recently been defined as a live microbial feed supplement that improves the health of IBS patients [61–63]. This microbe might also be beneficial if the *Strongyloides* infection is asymptomatic. Bacterial species typical of healthy people, such as *Desulfovibrio piger*, *Bifidobacterium bifidum*, *Enterococcus durans* and *Streptococcus salivarius*, were most prevalent in the NegST

group. Unexpectedly, the species *Escherichia coli* and *Klebsiella pneumoniae* were most abundant in the SsST group. Co-infection of these organisms with helminths and protozoa may be responsible for an increase in disease virulence [64, 65].

The difference between sample groups from the northeastern and the southern regions was statistically significant in each non-infected and infected group comparison at the species level. *Tetragenococcus halophilus* was the most common species in the northeastern group, with higher abundance in the SsNE group than in SsST, and in the NegNE group than the NegST group. This halophilic lactic-acid bacterium is found in a variety of fermented foods, such as soy sauce, shrimp paste and salted fish sauce [66, 67]. *Bradyrhizobium* sp. was more abundant in the SsST than in the SsNE group. This is an environmental bacterium capable of degrading aromatic compounds and of nitrogen fixation [68]. The difference in the gut microbiome pattern of the two regions that we studied could be due to a variety of factors such as dietary habits, consumption of probiotics and prebiotics, antibiotic use, intestinal comorbidities, and even metabolic diseases, all of which may continuously alter microbiota diversity and function, which is unique in each community [69–71].

Since many factors in the environment can affect microbial diversity, alpha and beta diversity metrics are required to quantify differences [72, 73]. However, we detected no significant differences in the overall community structure according to alpha indices, and some dissimilarity between microbial communities according to beta indices (Fig 3). The alpha-diversity results were consistent with those of studies on effects of soil-transmitted parasites on gut microbiota, which found no evidence of a difference between infected and uninfected groups [7, 12]. This observation differs from a previous study of the effect of *S. stercoralis* on the fecal microbiota of a cohort of human volunteers from a non-endemic area of northern Italy [39]. Various studies have suggested that the infection stage at which samples were collected may affect bacterial diversity, as acute parasite infection did not result in significant alterations of the fecal microbiota [74]. Furthermore, the baseline composition of the gut microbiota of individuals, as well as the parasite species involved, might be linked to observable alterations in the microbial community [12, 74, 75]. Our data show that, although *S. stercoralis* infection leads to minor changes in relative abundance of individual bacterial species, no major effect is observed on community structure and diversity. Prospective readers should be aware, however, that only a small number (N = 20) of human fecal samples were investigated: additional strongyloidiasis and healthy cases are required to confirm differences in microbial diversity.

Bacterial diversity may be influenced by geography and demographic variances. Indeed, the differences between *S. stercoralis*-infected and uninfected individuals were not dramatic, suggesting that a larger sample size for each group is required to corroborate our findings. However, we believe it is possible that differences in dietary habits in the two different regions of Thailand may have caused the observed differences in the composition of the gut microbiota. Furthermore, three SsST samples were obtained on different platforms, despite previous reports indicating that both platforms (Illumina and Ion Torrent) provided nearly identical results in terms of community composition and were able to reflect that the difference in relative abundance between samples is basically similar [76]. To improve reproducibility and reliability in the future, employing the same sequencing platforms or running the same sample in parallel across multiple platforms to reduce systematic error in microbiome analysis need to be done.

## Supporting information

**S1 Fig. Between-group variation analysis of selected species using *t*-tests.**
(PDF)

**S1 Table. Number of reads and OTUs for each sample.**
(XLSX)

**S2 Table. Complete list of bacterial OTUs found across 4 sample groups.**
(XLSX)

**S3 Table. List of unique, shared and core OTUs in each sample group.**
(XLSX)

**S4 Table. Statistical analysis of multiple factor comparison between groups at the genus level using two-way ANOVA.**
(XLSX)

## Acknowledgments

We would like to thank Professor David Blair through Khon Kaen University Publication Clinic (Thailand) for the English editing of this manuscript.

## Author Contributions

**Conceptualization:** Rutchanee Rodpai, Oranuch Sanpool, Penchom Janwan, Patcharaporn Boonroumkaew, Lakkhana Sadaow, Tongjit Thanchomnang, Pewpan M. Intapan, Wanchai Maleewong.

**Data curation:** Rutchanee Rodpai, Oranuch Sanpool, Tongjit Thanchomnang, Wanchai Maleewong.

**Formal analysis:** Rutchanee Rodpai, Oranuch Sanpool, Penchom Janwan, Patcharaporn Boonroumkaew, Lakkhana Sadaow, Tongjit Thanchomnang.

**Funding acquisition:** Rutchanee Rodpai, Oranuch Sanpool, Pewpan M. Intapan, Wanchai Maleewong.

**Investigation:** Rutchanee Rodpai, Oranuch Sanpool, Penchom Janwan, Patcharaporn Boonroumkaew, Lakkhana Sadaow, Tongjit Thanchomnang.

**Methodology:** Rutchanee Rodpai, Oranuch Sanpool, Penchom Janwan, Patcharaporn Boonroumkaew, Lakkhana Sadaow, Tongjit Thanchomnang, Pewpan M. Intapan, Wanchai Maleewong.

**Project administration:** Wanchai Maleewong.

**Supervision:** Pewpan M. Intapan, Wanchai Maleewong.

**Writing – original draft:** Rutchanee Rodpai, Wanchai Maleewong.

**Writing – review & editing:** Rutchanee Rodpai, Oranuch Sanpool, Penchom Janwan, Patcharaporn Boonroumkaew, Lakkhana Sadaow, Tongjit Thanchomnang, Pewpan M. Intapan, Wanchai Maleewong.

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
