## [Decision Letter · Decision Letter 0]

13 Jul 2022

PONE-D-22-05138Gut microbiota diversity in patients with strongyloidiasis differs little in two different regions in endemic areas of ThailandPLOS ONE

Dear Dr. Maleewong,

Thank you for submitting your manuscript to PLOS ONE. After careful consideration, we feel that it has merit but does not fully meet PLOS ONE’s publication criteria as it currently stands. Therefore, we invite you to submit a revised version of the manuscript that addresses the points raised during the review process. Should the authors address the points raised by the Reviewers. Specially regarding the statistics approaches used in the study.

We look forward to receiving your revised manuscript.

Kind regards,

Marcello Otake Sato, Ph.D., D.V.M.

Academic Editor

PLOS ONE

Journal Requirements:

Reviewers' comments:

Reviewer's Responses to Questions

**Comments to the Author**

1. Is the manuscript technically sound, and do the data support the conclusions?

Reviewer #1: Partly

Reviewer #2: Yes

2. Has the statistical analysis been performed appropriately and rigorously? 

Reviewer #1: No

Reviewer #2: Yes

3. Have the authors made all data underlying the findings in their manuscript fully available?

Reviewer #1: Yes

Reviewer #2: Yes

4. Is the manuscript presented in an intelligible fashion and written in standard English?

Reviewer #1: Yes

Reviewer #2: Yes

5. Review Comments to the Author

Reviewer #1: Summary

The manuscriot “Gut microbiota diversity in patients with strongylodiasis differs little in two different regions in endemic area of Thailand” by Maleewong et. al. surveyed gut microbiome from 5 individuals from two different regions and either infected or uninfected with intestinal nematode S. stercoralis. Authors report no significant difference as a function on infection, but observed differences in microbiomes of uninfected individual from different regions.

Major Concerns----------

I feel that the authors’ utilization of statistical analysis needs to be reconsidered to make the most of the data they have collected. A t-test was used for comparing sample types. The author’s experimental design however is highly appropriate to a two-factor Analysis of Variance 2-ANOVA (factors = NE/ST and infected/uninfected). I strongly suggest authors perform this or similar analysis, and perhaps additional statistical significance in the data may be uncovered.

Throughout manuscript, but particularly apparent for example in Venn-diagram (Fig 2), it is unclear how Authors combined experimental data from individuals. For example, in Fig 2 there are 1403 OTUs unique to SsST. Are those OTUs present in at least one individual from SsST? All SsST individuals? This should be made clear in analysis. I would additionally suggest, if authors have not already done so, remove OTUs found only in one or in very few individuals to remove very low prevalence/abundance OTUs that are on the edge of detectability or may even be sequencing artifacts.

Throughout the manuscript, but for example L39-41, Authors report changes in abundance that do not reach the level of statistical significance. I would state that there is no such thing as a ‘non-statistically significant change in abundance’; a change is statistically significant or else there is no change. I strongly recommend removing all reporting of non-significant results.

Two sequencing methods were reported as used in this study: Illumina and IonS5. These sequencing methods each have their own specific sequencing biases. While I do not think that mixing data from different sequencing technologies is inherently incompatible, Authors should, at the very least, identify which samples derive from which method, and ideally should present some analysis indicating that sequencer technology did not contribute to biases in OTU reporting.

Infection positive individuals are referred to in the manuscript as “patients”. What can Authors report about selection of infected and non-infected individuals? Are non-infected individuals also ‘pateints’? If possible, Authors should present some minimal aggregate demographic information for experimental groups (e.g. range of ages, percent male/female, or presence of co-morbidities for hospitalized patients).

Authors did not find significant differences in alpha or beta diversity as a function of strongylodiasis infection. The experimental sample size per experimental group however was relatively modest (N=5). The effects of sample size on analysis should be considered in Discussion section.

In the Introduction, Authors report “diverse and contrasting results” from prior investigations into the effect of parasites on host microbiomes. This requires elaboration. One significant frustration for me in reading this paper was lack of context. A clear description of what is known or suspected about the interactions between parasite infections and microbiomes, what specific question this study seeks to address, where the results of this study fall into the spectrum of previously reported 'diverse and contrasting results', and what makes this study unique relative to previous studies is needed.

Minor Concerns----------

In my opinion, the Abstract is too detailed. It would serve better by simply stating results in prose and saving details like p-values for Results section.

I cannot see where the number of samples analyzed is explicitly stated in manuscript. (N=5 for each condition, but only inferred that from supplemental data table). Number of samples should be in Methods (and if I missed it somehow, then perhaps it should be stated more prominently).

Discussion includes a number of Author-picked examples of taxa from analysis with a brief discussion of its possible role in community. The discussion may be stronger if some of the more tenuous links to possible functions are omitted and say nothing about a particular taxon rather than speculate too freely.

Figure 4: I don’t think that Figure 4 contributes much to description of data. Maybe an alternative version of this figure or just prose description of data in main text.

In Discussion (L235-236), “It has been suggested that the gut microbiota influences human health”. I suggest that it is safe to call the link between microbiome and human health well-established at this time.

Recommendation----------

Major Revision

Reviewer #2: 6. Review Comments to the Author (minimum 200 characters)

The manuscript entitled “Gut microbiota diversity in patients with strongyloidiasis differs little in two different regions in endemic areas of Thailand” by Rodpai et al, showed that S. stercoralis infection leads to only minor alterations in the relative abundance of individual bacterial species in the human gut, and no detectable effect was observed on community structure and diversity. Some differences in species abundance were noted between geographical isolates. There are several points concerning clarity in the manuscript.

- line 72-74: S. stercoralis infection prevalence in 4 regions of Thailand. Explain more why the study area focused only in the southern and the northeastern areas and only one province from each area was selected.

- Please give more information about the sample size. Only 10 stool samples (5 SS positive and 5 SS negative) from each province were used for microbial composition analysis.

- line 85- amount of fecal samples and number of volunteers should be added

- Diet, age and health condition play an important role on microbial gut composition, therefore basic information of the volunteers, for examples age, health condition, diet, should be added.

-line 89- A portion means how many grams of each fecal sample

-line 91 The stool samples were categorized in 4 groups. Give more detail about DNA preparation of stool samples from each area. Is it prepared separately, or are pool samples from each area together and then extracted for the DNA?

Discussion

Line 243- please discuss more about the Ruminococcaceae abundance in uninfected groups and low abundance in infected groups.

6. PLOS authors have the option to publish the peer review history of their article (what does this mean?). If published, this will include your full peer review and any attached files.

Reviewer #1: No

Reviewer #2: **Yes: **Wilawan Pumidonming

---

## [Author Response · Author response to Decision Letter 0]

23 Aug 2022

Academic Editor

Journal Requirements:

Reply: We have done as suggested.

Reply: We have done as suggested. We provided the correct grant numbers of our grants in the ‘Funding Information’ section and cover letter.

Reply: We have done as suggested. All sequence reads have been deposited at the NCBI Sequence Read Archive (SRA) under project accession number PRJNA807511. 

https://www.ncbi.nlm.nih.gov/bioproject/PRJNA807511

Reviewer #1: Summary

The manuscript “Gut microbiota diversity in patients with strongylodiasis differs little in two different regions in endemic area of Thailand” by Maleewong et. al. surveyed gut microbiome from 5 individuals from two different regions and either infected or uninfected with intestinal nematode S. stercoralis. Authors report no significant difference as a function on infection, but observed differences in microbiomes of uninfected individual from different regions.

Major Concerns----------

I feel that the authors’ utilization of statistical analysis needs to be reconsidered to make the most of the data they have collected. A t-test was used for comparing sample types. The author’s experimental design however is highly appropriate to a two-factor Analysis of Variance 2-ANOVA (factors = NE/ST and infected/uninfected). I strongly suggest authors perform this or similar analysis, and perhaps additional statistical significance in the data may be uncovered.

Reply: We have more analyzed as suggestion. Please see in materials and methods section (lines 148-150, page 7), and in results section (lines 236-245, pages 10-11) and S4 Table.

Throughout manuscript, but particularly apparent for example in Venn-diagram (Fig 2), it is unclear how Authors combined experimental data from individuals. For example, in Fig 2 there are 1403 OTUs unique to SsST. Are those OTUs present in at least one individual from SsST? All SsST individuals? This should be made clear in analysis. I would additionally suggest, if authors have not already done so, remove OTUs found only in one or in very few individuals to remove very low prevalence/abundance OTUs that are on the edge of detectability or may even be sequencing artifacts.

Reply: We have added the sentence for more clear in OTU clustering analysis, please see lines 137-139, page 6. Because all sequences or abundances of each OTU were utilized for analysis, your suggestions to exclude OTUs detected only in one sample or in very low abundances, the results will affect on the alpha indices (even have a singleton sequence). If these were removed, the outcome would be biased. However, we have added the OTUs result of individuals in the S2 Table (please see revised S2 Table).

Throughout the manuscript, but for example L39-41, Authors report changes in abundance that do not reach the level of statistical significance. I would state that there is no such thing as a ‘non-statistically significant change in abundance’; a change is statistically significant or else there is no change. I strongly recommend removing all reporting of non-significant results.

Reply: We have deleted the sentence throughout the text to address these points.

Two sequencing methods were reported as used in this study: Illumina and IonS5. These sequencing methods each have their own specific sequencing biases. While I do not think that mixing data from different sequencing technologies is inherently incompatible, Authors should, at the very least, identify which samples derive from which method, and ideally should present some analysis indicating that sequencer technology did not contribute to biases in OTU reporting.

Reply: We agree with your suggestion. We have added more information on which samples derive from which method (Please see lines 121-122, pages 5-6), and we added in discussion section, please see lines 348-354, page 15.

Infection positive individuals are referred to in the manuscript as “patients”. What can Authors report about selection of infected and non-infected individuals? Are non-infected individuals also ‘pateints’? If possible, Authors should present some minimal aggregate demographic information for experimental groups (e.g. range of ages, percent male/female, or presence of co-morbidities for hospitalized patients).

Reply: We have modified as suggestion, please see in materials and methods section (lines 84-99, pages 4-5).

Authors did not find significant differences in alpha or beta diversity as a function of strongylodiasis infection. The experimental sample size per experimental group however was relatively modest (N=5). The effects of sample size on analysis should be considered in Discussion section.

Reply: We have more discussed in the discussion section. Please see lines 340-342, page 14, and lines 345-346, page 15.

In the Introduction, Authors report “diverse and contrasting results” from prior investigations into the effect of parasites on host microbiomes. This requires elaboration. One significant frustration for me in reading this paper was lack of context. A clear description of what is known or suspected about the interactions between parasite infections and microbiomes, what specific question this study seeks to address, where the results of this study fall into the spectrum of previously reported 'diverse and contrasting results', and what makes this study unique relative to previous studies is needed.

Reply: We have added more information in the introduction part, please see lines 66-74. The previous study suggested that the parasitic adult stages of S. stercoralis inhabit the GI tract of humans, where their interactions with the host gut microbiota can impact, and may alter, the host gut microbiome. Furthermore, race and geographic location may influence the composition and diversity of the gut microbiota. The aim of this study was investigated the effect of S. stercoralis infection on the composition of human gut microbiota and whether such effects differed between different areas, the southern and the northeastern region of Thailand, which is differed the most in terms of geography, population behavior and diets. Also, we added sentences in the discussion part, please see lines 325-333.

Minor Concerns----------

In my opinion, the Abstract is too detailed. It would serve better by simply stating results in prose and saving details like p-values for Results section.

Reply: We have deleted as suggested. Please see in Abstract section., revised version. 

I cannot see where the number of samples analyzed is explicitly stated in manuscript. (N=5 for each condition, but only inferred that from supplemental data table). Number of samples should be in Methods (and if I missed it somehow, then perhaps it should be stated more prominently).

Reply: We have modified as suggestion, please see in materials and methods section (line 95-99, pages 4- 5).

Discussion includes a number of Author-picked examples of taxa from analysis with a brief discussion of its possible role in community. The discussion may be stronger if some of the more tenuous links to possible functions are omitted and say nothing about a particular taxon rather than speculate too freely.

Reply: Thank you for your insightful remark. The aim that we discussed are the main significant taxon that possible impact to Strongyloides infection in different endemic regions provided the information to the academic readers who works in this field. Also, for better understanding, we modified some sentences in thoroughly discussion part, please see revised version.

Figure 4: I don’t think that Figure 4 contributes much to description of data. Maybe an alternative version of this figure or just prose description of data in main text.

Reply: We have removed the figure 4 to S1 Fig. and described of data in the text. Please see lines 228-236, page 10.

In Discussion (L235-236), “It has been suggested that the gut microbiota influences human health”. I suggest that it is safe to call the link between microbiome and human health well-established at this time.

Reply: We have modified the sentence as suggestion, please see line 249, page 11.

Recommendation----------

Major Revision

Finally, we would like to thank the reviewer very much for your kind suggestions, your comments have supportive and helpful. 

Reviewer #2: 

The manuscript entitled “Gut microbiota diversity in patients with strongyloidiasis differs little in two different regions in endemic areas of Thailand” by Rodpai et al, showed that S. stercoralis infection leads to only minor alterations in the relative abundance of individual bacterial species in the human gut, and no detectable effect was observed on community structure and diversity. Some differences in species abundance were noted between geographical isolates. There are several points concerning clarity in the manuscript.

- line 72-74: S. stercoralis infection prevalence in 4 regions of Thailand. Explain more why the study area focused only in the southern and the northeastern areas and only one province from each area was selected.

Reply: We added more detail to explain, please see in introduction section (lines 68-72, page 3).

- Please give more information about the sample size. Only 10 stool samples (5 SS positive and 5 SS negative) from each province were used for microbial composition analysis.

Reply: We added more detail in number of sample size, please see in materials and methods section (lines 95-99, pages 4-5).

- line 85- amount of fecal samples and number of volunteers should be added

Reply: We have added, please see in Materials and methods section (line 85, page 4).

- Diet, age and health condition play an important role on microbial gut composition, therefore basic information of the volunteers, for examples age, health condition, diet, should be added.

Reply: We have added more information. please see in Materials and methods section (lines 87-99, pages 4-5).

-line 89- A portion means how many grams of each fecal sample

Reply: We have modified the sentence. Please see line 91, page 4

-line 91 The stool samples were categorized in 4 groups. Give more detail about DNA preparation of stool samples from each area. Is it prepared separately, or are pool samples from each area together and then extracted for the DNA?

Reply: We have modified the text. Please see in Materials and methods section (line 103-115, page 5). Each stool sample was prepared individually and the DNA sample was extracted separately (did not use pooled samples).

Discussion

Line 243- please discuss more about the Ruminococcaceae abundance in uninfected groups and low abundance in infected groups.

Reply: We have more discussed in the discussion section. Please see lines 259-264, page 11.

Finally, we appreciate the reviewer very much for your kind suggestions, your comments have supportive and helpful.

---

## [Decision Letter · Decision Letter 1]

14 Dec 2022

Gut microbiota diversity in human strongyloidiasis differs little in two different regions in endemic areas of Thailand

PONE-D-22-05138R1

Dear Dr. Maleewong,

We’re pleased to inform you that your manuscript has been judged scientifically suitable for publication and will be formally accepted for publication once it meets all outstanding technical requirements.

Kind regards,

Marcello Otake Sato, Ph.D., D.V.M.

Academic Editor

PLOS ONE

Additional Editor Comments (optional):

Reviewers' comments:

Reviewer's Responses to Questions

**Comments to the Author**

1. If the authors have adequately addressed your comments raised in a previous round of review and you feel that this manuscript is now acceptable for publication, you may indicate that here to bypass the “Comments to the Author” section, enter your conflict of interest statement in the “Confidential to Editor” section, and submit your "Accept" recommendation.

Reviewer #2: All comments have been addressed

Reviewer #3: All comments have been addressed

2. Is the manuscript technically sound, and do the data support the conclusions?

Reviewer #2: Yes

Reviewer #3: Yes

3. Has the statistical analysis been performed appropriately and rigorously? 

Reviewer #2: Yes

Reviewer #3: Yes

4. Have the authors made all data underlying the findings in their manuscript fully available?

Reviewer #2: Yes

Reviewer #3: Yes

5. Is the manuscript presented in an intelligible fashion and written in standard English?

Reviewer #2: Yes

Reviewer #3: (No Response)

6. Review Comments to the Author

Reviewer #2: The authors have adequately addressed to comments mentioned in a previous round of review.

The data support the conclusions. The statistical analysis been performed appropriately. This manuscript is now acceptable for publication.

Reviewer #3: (No Response)

7. PLOS authors have the option to publish the peer review history of their article (what does this mean?). If published, this will include your full peer review and any attached files.

Reviewer #2: No

Reviewer #3: **Yes: **Ana Julia Pinto Fonseca Sieuve Afonso

---

## [Editor Report · Acceptance letter]

21 Dec 2022

PONE-D-22-05138R1 

Gut microbiota diversity in human strongyloidiasis differs little in two different regions in endemic areas of Thailand  

Dear Dr. Maleewong:

I'm pleased to inform you that your manuscript has been deemed suitable for publication in PLOS ONE. Congratulations! Your manuscript is now with our production department. 

Kind regards, 

on behalf of

Dr. Marcello Otake Sato 

Academic Editor

PLOS ONE